# Crowdfunding for complementary and alternative medicine: What are cancer patients seeking?

**Jeremy Snyder** [1]*, **Marco Zenone**[2], **Timothy Caulfield**[3]

1 Faculty of Health Sciences, Simon Fraser University, Burnaby, British Columbia, Canada, 2 Faculty of Public Health and Policy, London School of Hygiene and Tropical Medicine, London, United Kingdom, 3 Health Law Institute, University of Alberta, Edmonton, Alberta, Canada

* jcs12@sfu.ca

**Data Availability Statement:** The data is available at figshare: https://figshare.com/articles/dataset/CAM_Cancer_Data_xlsx/13204022.

**Funding:** The authors received funding from the Greenwall Foundation that supported this work.

## Abstract

### Background

Complementary and alternative medicine (CAM) is increasingly being integrated into conventional medical care for cancer, used to counter the side effects of conventional cancer treatment, and offered as an alternative to conventional cancer care. Our aim is to gain a broader understanding of trends in CAM interventions for cancer and crowdfunding campaigns for these interventions.

### Methods

GoFundMe campaigns fundraising for CAM were retrieved through a database of crowdfunding campaign data. Search terms were drawn from two National Institutes of Health lists of CAM cancer interventions and a previous study. Campaigns were excluded that did not match these or related search terms or were initiated outside of June 4th, 2018 to June 4th, 2019.

### Results

1,396 campaigns were identified from the US (n = 1,037, 73.9%), Canada (n = 165, 11.8%), and the UK (n = 107, 7.7%). Most common cancer types were breast (n = 344, 24.6%), colorectal (n = 131, 9.4%), and brain (n = 98, 7.0%). CAM interventions sought included supplements (n = 422, 30.2%), better nutrition (n = 293, 21.0%), high dose vitamin C (n = 276, 19.8%), naturopathy (n = 226, 16.2%), and cannabis products (n = 211, 15.1%). Mexico (n = 198, 41.9%), and the US (n = 169, 35.7%) were the most common treatment destinations.

### Conclusions

These findings confirm active and ongoing interest in using crowdfunding platforms to finance CAM cancer interventions. They confirm previous findings that CAM users with cancer tend to have late stage cancers, cancers with high mortality rates, and specific diseases

**Competing interests:** The authors have declared that no competing interests exist.

such as breast cancer. These findings can inform targeted responses where facilities engage in misleading marketing practices and the efficacy of interventions is unproven.

## Introduction

Complementary and alternative medicine (CAM) is made up of medical products and practices that are not part of conventional medical practice and care [1, 2]. However, CAM is increasingly being integrated into conventional medical care for cancer, used to counter the side effects of conventional cancer treatment, and offered as an alternative to conventional cancer care [3]. Despite the growing popularity, the studies on the safety and efficacy of CAM cancer interventions are mixed and have tended to be of poor quality [4, 5]. At the same time, use of CAM cancer treatments is associated with significant potential harms [6]. In some cases, CAM treatments can interfere with the functioning of conventional treatments [3]. People seeking CAM cancer treatments are also more likely to refuse conventional cancer treatment [7]. This is concerning as delaying or refusing to use conventional cancer treatments in favour of CAM treatments can reduce cancer survival rates [8].

In addition to potentially reducing the survivability of cancer, use of CAM can harm other aspects of wellbeing. When CAM interventions are ineffective or less effective than expected, they can lead to significant financial costs to users. CAM modalities for cancer treatment are often paid for out-of-pocket and add up to $445 per person utilizing them or $6.7 billion annually in the United States (US) [9]. Moreover, misinformation about the safety and efficacy of CAM treatments for cancer is common and can create false hope about the likely effects of these interventions [10]. Despite these concerns, some view the increased use of CAM for cancer as a result of the weaknesses of conventional cancer care, including an unjustified toleration of misinformation by some CAM providers, poor management of the negative side effects of conventional treatment, inadequate access to palliative care, and the patriarchal structure of conventional medicine [11, 12].

There are significant challenges to understanding trends in how CAM is used due to scarcity of data and hesitancy of patients to disclose some CAM use to their physicians [8]. Studies in the US have found that 79% of cancer survivors used a CAM modality in the previous year, most commonly vitamins and minerals (74.8%), non-vitamin and mineral natural products (23.7%), manipulative and body-based therapies including chiropracty and massage (18.6%), homeopathy (2.9%), acupuncture (2.0%), naturopathy (1.0%), and energy healing (0.9%) [9]. Other studies have found the mean percentage of people with cancer using CAM globally to be 51% [3]. Those using alternative treatments in lieu of conventional treatment are more likely to be female, at a higher cancer stage, young, better educated, and wealthier than the general population of people with cancer [3, 8, 9, 13, 14]. People with breast, melanoma, or colorectal cancer have been found to be more likely to seek CAM than those with prostate cancer [7, 9].

Because CAM cancer treatments are often not paid for by public or private insurance, individuals desiring to pursue these treatments can face significant out of pocket costs. As a result, a growing number of people seek financial support from friends, family, and even strangers through online crowdfunding, including for CAM cancer treatments [15, 16]. These campaigns include user-generated accounts of their cancer diagnosis, CAM treatments sought, and providers of CAM interventions. For example, one study of crowdfunding campaigns for people with cancer seeking homeopathic treatments found that these recipients also sought CAM treatments including food and diet changes, natural supplements, vitamins and

minerals, oxygen and ozone therapies, cannabis-based treatments, energy healing, and hyper-thermia [17]. As such, these campaigns have the potential to offer valuable and timely informa-tion on trends in CAM cancer interventions. This data can also provide an understanding of CAM usage by people with cancer that complements existing data obtained from physicians and in a clinical setting.

Many recipients of crowdfunding campaigns for CAM interventions choose to forgo con-ventional treatment or palliative care. These recipients are often very ill, as demonstrated in one study where at least 28% of recipients had died following the start of their campaign [17]. The use of crowdfunding for CAM cancer interventions also raises distinct concerns because these campaigners may seek CAM treatments with little to no evidentiary support, directing money from large pools of people to clinics with problematic business practices. Previous analyses of these campaigns have flagged the Burzynski clinic in Texas, the Hallwang Private Oncology Clinic in Germany, and providers selling the Gerson therapy as a cancer treatment as engaging in misleading marketing or selling products that may put patients at risk [15]. Moreover, studies of crowdfunding campaigns for alternative or unproven interventions have found that these campaigners often repeat and exaggerate misinformation about the safety and efficacy of these interventions, use markers of scientific legitimacy to support their campaigns, and are used to fund ineffective and potentially dangerous interventions [18–21]. This is not surprising given that these campaigns must reassure potential donors of the value of these interventions and that their money will be well spent. But in doing so, these campaigns poten-tially spread misinformation about CAM and unproven treatments to a large audience.

Thus, it is important to gain a broader understanding of the dimensions of crowdfunding campaigns for CAM cancer treatments both as an insight into developing trends in demand for these interventions generally and to better understand their scope in crowdfunding cam-paigns specifically.

## Methods

The GoFundMe crowdfunding platform was selected to identify crowdfunding campaigns for CAM cancer interventions as this is by far the largest host of medical crowdfunding campaigns globally [22]. GoFundMe.com campaigns fundraising for CAM treatments were retrieved through a database of scraped crowdfunding campaign data that records all campaign text but does not include images and video. This database is a searchable collection of all GoFundMe. com campaigns on the website that began recording campaign data in April, 2019. Ethics approval was not required for the use of this data as it is publicly available without an expecta-tion of privacy. Recorded data includes the campaign title, amount pledged, amount requested, campaign type category, fundraiser location, Facebook shares, campaign description, and updates. Each search conducted on the portal is exported as a CSV file.

To identify search terms for CAM cancer campaigns, we utilized two National Institutes of Health (NIH) lists of CAM cancer treatments and treatment types and a list of CAM treatment types identified in a previous study of crowdfunding for CAM cancer interventions [17, 23, 24]. From these sources, 123 CAM interventions were identified. After identification of these search terms, the authors reviewed the included search terms and recommended removing CAM treatments that are common foods, such as vegetables, or that were health oriented behaviours that were too broadly framed to be considered as solely alternative in nature, such as exercise. The authors additionally identified similar terms and labels for each identified CAM treatment to identify alternative language used by crowdfunders to fundraise (for exam-ple, cannabis synonyms include marijuana and cannabidiol). After the authors agreed upon the search terms, 110 CAM treatments and their identified related terms were searched on the

database with the word "cancer" (see S1 Appendix). The search was conducted from June 9 – July 25, 2019 and identified campaigns for each individual search were recorded on individual spreadsheets.

These searches yielded a total of 16,506 campaigns. After duplicate campaigns were removed, there were 10,619 campaigns. Campaign categories irrelevant to those funding for medical purposes were removed, including: animals, business, California fire, Canada 150, competitions, creative, education, Hurricane Maria, memorials, Nepal, newlyweds, sports, volunteer, and wishes. The campaign categories included, which are applicable to those fundraising for medical reasons, are: cancer, charity, community, emergencies, events, faith, family, medical, other, travel, and blank. The exclusion of campaign categories irrelevant to medical purposes reduced the number of campaigns to 9,326.

A one-year campaign launch time range of June 4th, 2018 to June 4th, 2019 was implemented to create a dataset of a manageable size, leaving 2,904 campaigns. These campaigns were organized in a spreadsheet and split for review between the first and second author. Campaigns were included only if a CAM intervention matching the search terms or related to these terms was sought for cancer. The third author reviewed 5% of these campaigns during the review process to verify consistent application of inclusion and exclusion criteria and disagreements were discussed and resolved by the authors. A total of 1,396 campaigns met this inclusion criterion and had their campaign characteristics recorded and tabulated. Recorded data included the stage of cancer, cancer type, CAM treatment(s), provider('s) location(s), and provider('s) name(s).

## Results

These 1,396 campaigns were supported by 122,701 (median 49) donors and shared on Facebook 577,351 (median 234) times. They requested $39,611,973.20 (median $19,880) and were pledged $12,756,563.92 (median $5,055.50) or 32.2% of that amount. 1,037 (73.9%) of these campaigners were from the US, 165 (11.8%) from Canada, 107 (7.7%) from the UK, and 57 (4.1%) from Australia. Of the remaining campaigns, 28 (2%) were from Europe, 2 from Japan, and 1 each from Costa Rica, the Dominican Republic, Mauritius, Panama, and the Philippines (see Table 1) (see Figs 1–3). When recipients' cancer stage was described, these skewed toward late stage cancers, including stage 4 (n = 454, 55.8%), stage 3 (n = 130, 16.0%), and cancers described as metastatic or late stage (n = 102, 12.5%) or terminal or incurable (n = 79, 9.7%). The remaining campaigns identified stage 2 (n = 42, 5.2%) and stage 1 (n = 7, 0.9%) cancer diagnoses (see Table 2).

The recipients of these campaigns were described as having a broad range of cancer types. By far the most common of these was breast cancer (n = 344, 24.6%), followed by unspecified (n = 138, 9.9%), colorectal (n = 131, 9.4%), brain (n = 98, 7.0%), lung (n = 84, 6.0%), pancreatic (n = 61, 4.4%), gastrointestinal (n = 55, 3.9%), ovarian (n = 54, 3.9%), cervical (n = 42, 3.0%), prostate (n = 34, 2.4%), lymphomas excluding non-Hodgkin (n = 30, 2.2%), and liver (n = 28, 2.0%) (see Table 3). This distribution of cancer types in some cases broadly tracks with US mortality rates, including colorectal cancers, gastrointestinal cancers, lymphomas, and melanomas. In other cases, US mortality rates were much higher than found in these campaigns, including lung cancer, pancreatic cancers, liver cancer, leukemia, and prostate cancer. Breast, brain, ovarian, and cervical cancer were much more common in these campaigns than incidence and mortality rates in the US and studies of crowdfunding campaigns found (see Table 4).

The most common CAM interventions sought in these campaigns were a range of dietary supplements (n = 422, 30.2%). These supplements were described both in general terms and as

**Table 1. Campaigner locations.**

| Campaigner Location | Number of Campaigns | Percentage of Campaigns |
|---|---|---|
| United States | 1032 | 73.92% |
| Canada | 165 | 11.82% |
| United Kingdom | 107 | 7.66% |
| Australia | 57 | 4.08% |
| Germany | 8 | 0.57 |
| Ireland | 8 | 0.57 |
| Spain | 4 | 0.29 |
| Italy | 2 | 0.14 |
| Japan | 2 | 0.14 |
| Switzerland | 2 | 0.14 |
| Costa Rica | 1 | 0.07 |
| Denmark | 1 | 0.07 |
| Dominican Republic | 1 | 0.07 |
| France | 1 | 0.07 |
| Mauritius | 1 | 0.07 |
| Netherlands | 1 | 0.07 |
| Panama | 1 | 0.07 |
| Philippines | 1 | 0.07 |
| Portugal | 1 | 0.07 |

specific lists of items, as in one campaign that sought: "a variety of supplements such as Vit C, D3, B17, oxygen, milk thistle for detoxing, among other more esoteric natural substances geared at killing cancer cells or boosting the immune system". Another common CAM intervention was healthy food, better nutrition, organic food, or changes to diet (n = 293, 21.0%) that were often justified in terms of supporting greater overall health or immunity. For

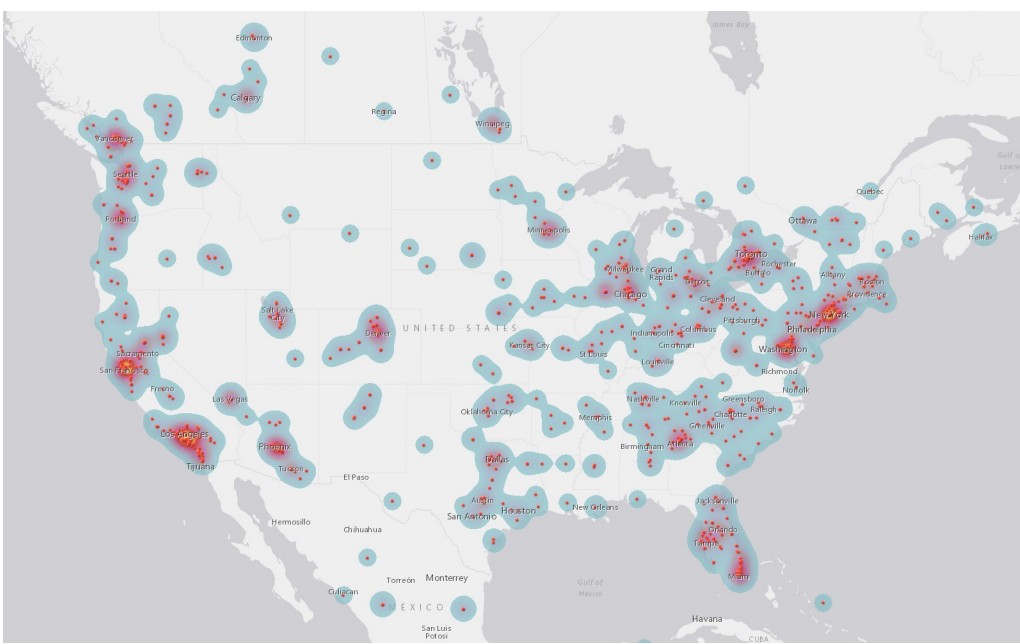

**Fig 1.** US campaigner locations.

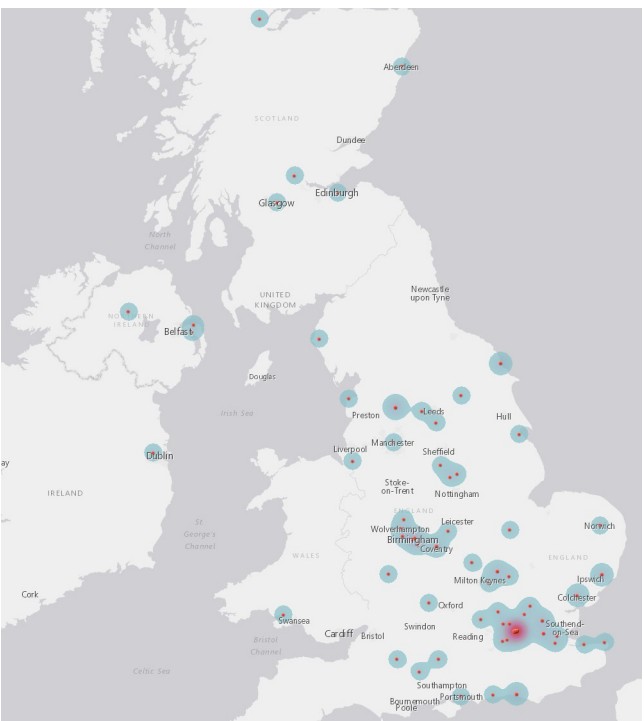

**Fig 2. UK and Ireland campaigner locations.**

example, one campaigner stated that "I believe we can heal all things with a healthy body . . . and sure want to go that route". Intravenous or high dose vitamin C (n = 276, 19.8%) appeared commonly and was said, among other things, to have the effect of "giving me more energy. Friends tell me my color is better". As with dietary supplements, naturopathic interventions (n = 226, 16.2%) were commonly praised as offering "holistic" care and in one case being

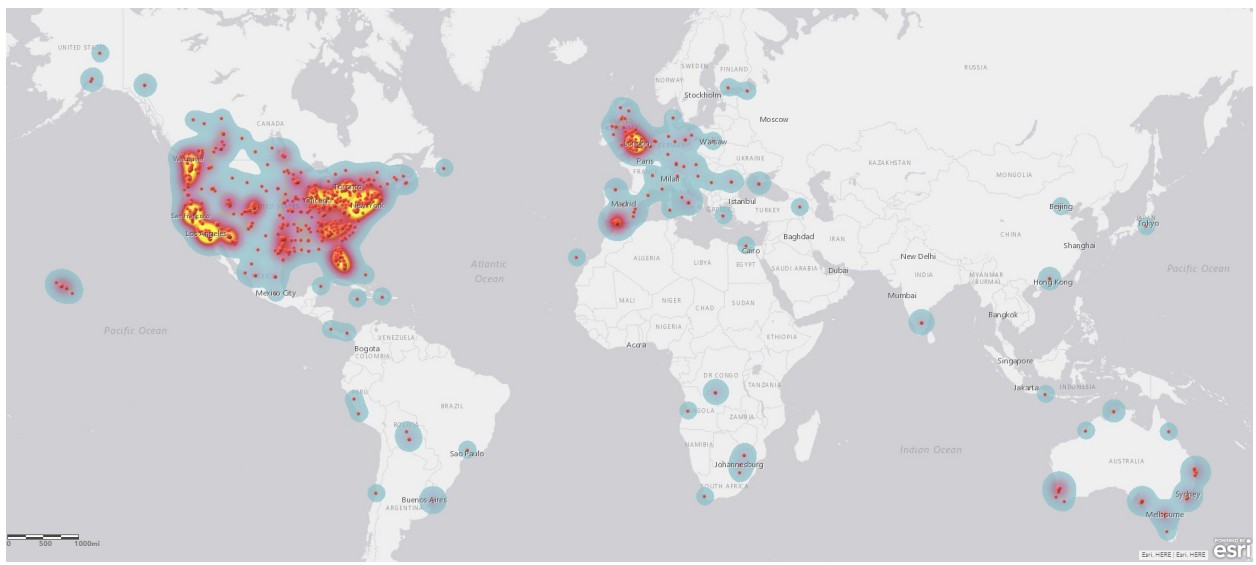

**Fig 3. Worldwide campaigner locations.**

**Table 2. Recipient cancer stage.**

| Cancer Stage | # | % Total | % Listed |
|---|---|---|---|
| Not Listed | 582 | 41.69 | N/A |
| 4 | 454 | 32.52 | 55.77 |
| 3 | 130 | 9.31 | 15.97 |
| Metastatic/Spread/Late Stage | 102 | 7.31 | 12.53 |
| Terminal/Incurable/Inoperable | 79 | 5.66 | 9.70 |
| 2 | 42 | 3.01 | 5.16 |
| 1 | 7 | 0.50 | 0.86 |

provided by "a great, supportive naturopathic doctor that specializes in oncology with very modern therapies from Europe". Cannabis products including cannabidiol (CBD) and Rick Simpson Oil (RSO) (n = 211, 15.1%) were frequently praised as offering more "natural" alternatives for pain relief and managing treatment side effects. This view is exemplified the campaigner who wrote that "They gave her morphine the first time which tells me that BIG PHARM has taken over so we are gonna try the Cbd route and maybe even marijuana route for appetite and putting on the weight". Other vitamins and minerals (n = 206, 14.8%) and herbs and mushrooms (n = 159, 11.4%) joined dietary supplements in being said to have immune and energy boosting properties.

Other commonly sought interventions included hyperbaric oxygen therapy (HBOT) and other oxygen interventions (n = 146, 10.5%), immune system boosting interventions (n = 123, 8.8%), acupuncture (n = 121, 8.7%), detoxification, purges, cleanses, and chelation (n = 111, 8.0%), hyperthermia and other heat therapies (n = 110, 7.9%), juicing (n = 96, 6.9%), ozone (n = 87, 6.2%), Gerson therapy (n = 82, 5.9%), and homeopathy (n = 76, 5.4%) (see Table 5). The decision to pursue these products was characteristically justified in terms of undertaking a "more natural and holistic approach" to cancer care; religious ideals such as "seeking alternative treatment, with a regimen of vitamins, jucing [sic] and whatever our Father God leads us to"; the desire to explore "every option to treat his cancer"; and a "last resort" after the failure of conventional treatment.

When these campaigns named treatment destinations and facilities, Mexico (n = 198, 41.9%), the US (n = 169, 35.7%), Germany (n = 37, 7.8%), and Canada (n = 19, 4.0%) were the most common destination countries (see Table 6) (see Figs 4 and 5). The most commonly named facility was CHIPSA Hospital in Tijuana, Mexico (n = 39, 11.0%), called "groundbreaking" and "the premier Gerson Therapy Center". Also located in Tijuana, Mexico, the Hope4Cancer Treatment Center (n = 33, 9.3%) was praised as "a wholly alternative, natural protocol institute running for the last 30 years, with a staggering 98% success rate of full recovery from stage 4 cancer". As the name would suggest, the Gerson Institute in Tijuana, Mexico and Budapest, Hungary (n = 23, 6.5%) was selected largely due to campaigners' desire to access the Gerson therapy. The Immunity Therapy Center in Tijuana, Mexico (n = 15, 4.2%) was praised as offering both integrative care and recognizing that there "are other choices beyond conventional cancer treatments when chemotherapy, radiation, and traditional medicine do not work". Another Tijuana, Mexico clinic, Oasis of Hope (n = 14, 3.9%), was selected for "a holistic atmosphere that addresses healing of the body, mind and spirit". As with campaigners seeking the Gerson therapy specifically, the campaigners seeking treatment at the Forsythe Cancer Care Center in Reno, Nevada (n = 11, 3.1%) sought a specific treatment regimen, as "Dr Forsythe's treatment shows results of about 80% tumor shrinkage" (see Table 7).

While there were regional differences in the destination preferences that were driven in part by geographic proximity, Mexico remained a global draw for campaigners. Among UK-

**Table 3. Cancer types and locations.**

| Cancer Type/Location | Number | Percentage |
|---|---|---|
| Breast | 344 | 24.64 |
| Unspecified | 138 | 9.89 |
| Colorectal | 131 | 9.38 |
| Brain | 98 | 7.02 |
| Lung | 84 | 6.02 |
| Pancreatic | 61 | 4.37 |
| Gastrointestinal | 55 | 3.94 |
| Ovarian | 54 | 3.87 |
| Cervical | 42 | 3.01 |
| Prostate | 34 | 2.44 |
| Lymphoma (including Hodgkin) | 30 | 2.15 |
| Liver | 28 | 2.01 |
| Soft Tissue Sarcoma | 27 | 1.93 |
| Leukemia | 26 | 1.86 |
| Uterine/Endometrial | 20 | 1.43 |
| Melanoma | 19 | 1.36 |
| Bladder | 18 | 1.29 |
| Non-Hodgkin's Lymphoma | 18 | 1.29 |
| Multiple Myeloma | 17 | 1.22 |
| Throat | 15 | 1.07 |
| Kidney | 14 | 1.00 |
| Squamous Cell | 14 | 1.00 |
| Bile Duct | 12 | .86 |
| Bone | 12 | .86 |
| Adenocarcinoma | 11 | .79 |
| Oral | 11 | .79 |
| Testicular | 10 | .72 |
| Thyroid | 10 | .72 |
| Neuroendocrine | 8 | .57 |
| Ewing's Sarcoma | 6 | .43 |
| Appendix | 5 | .36 |
| Sarcoma (Undefined) | 5 | .36 |
| Bone Marrow | 4 | .29 |
| Neck | 2 | .14 |
| Neuroblastoma | 2 | .14 |
| Vaginal | 2 | .14 |
| Peritoneal | 2 | .14 |
| Gallbladder | 1 | .07 |
| Mediastinal | 1 | .07 |
| Meningioma | 1 | .07 |
| Neurofibromatosis | 1 | .07 |
| Sinus | 1 | .07 |
| Thymic | 1 | .07 |
| Urethral | 1 | .07 |

based campaigns stating a destination, Germany (n = 9, 29.0%) was the most popular destination, followed by Mexico (n = 6, 19.6%), the UK (n = 5, 16.1%), Turkey (n = 3, 9.7%), Latvia

**Table 4. Cancer types by incidence and mortality.**

| Cancer Type | % of campaign recipients | % incidence (US)[1] | % mortality (US) | % cancer crowdfunding generally[2] |
|---|---|---|---|---|
| Breast | 24.64 | 15.39 | 6.96 | 18.3 |
| Unspecified | 9.89 | N/A | N/A | N/A |
| Colorectal | 9.38 | 9.33 | 8.88 | 7.0 |
| Brain | 7.02 | 1.35 | 2.93 | 4.2 |
| Lung | 6.02 | 12.95 | 23.51 | 10.9 |
| Pancreatic | 4.37 | 3.22 | 7.54 | 5.0 |
| Gastrointestinal | 3.94 | 2.56 | 4.49 | N/A |
| Ovarian | 3.87 | 1.28 | 2.30 | N/A |
| Cervical | 3.01 | 0.75 | 0.70 | N/A |
| Prostate | 2.44 | 9.91 | 5.21 | N/A |
| Lymphomas | 3.44 | 4.67 | 3.46 | N/A |
| Liver and Bile Duct | 2.87 | 2.38 | 5.24 | 4.2 |
| Soft Tissue Sarcoma | 1.93 | 0.72 | 0.87 | N/A |
| Leukemia | 1.86 | 3.51 | 3.76 | 13.0 |
| Uterine/Endometrial | 1.43 | 3.51 | 2.00 | N/A |
| Melanoma | 1.36 | 5.47 | 1.19 | N/A |
| Bladder | 1.29 | 4.57 | 2.91 | N/A |
| Multiple Myeloma | 1.22 | 1.82 | 2.14 | N/A |
| Kidney | 1.00 | 4.19 | 2.43 | N/A |
| Thyroid | .72 | 2.95 | 0.36 | N/A |

[1] https://www.cancer.org/content/dam/cancer-org/research/cancer-facts-and-statistics/annual-cancer-facts-and-figures/2019/cancer-facts-and-figures-2019.pdf
[2] https://escholarship.org/content/qt9b48t99p/qt9b48t99p.pdf

(n = 2, 6.5%), and 1 each for Canada, China, the Czech Republic, Poland, Spain, and Thailand. Australian-based campaigns sought interventions in Mexico (n = 6, 30.0%), Thailand (n = 4, 20.0%), Australia (n = 3, 15.0%), New Zealand (n = 2, 10.0%) and 1 each for Ecuador, Germany, Indonesia, Jamaica, and the US.

## Discussion

These findings confirm an ongoing and very active presence of crowdfunding campaigns for CAM cancer interventions on the GoFundMe platform. In some respects, the recipients of these campaigns overlap with and help support the findings of other studies of people seeking CAM interventions for cancer. Studies of these individuals have shown that they tend to have late stage cancers. This was the case in our findings as well, with 65.4% of those discussing their cancer stage describing themselves as having a stage 4 or terminal cancer diagnosis and only 6.0% having a stage 1 or 2 diagnosis.

The cancer types and locations in our findings commonly tracked more closely with mortality rates than incidence rates in the US, as with colorectal, gastrointestinal, lymphomas, melanoma, kidney, and thyroid cancers, though pancreatic and liver and bile duct cancers more closely tracked incidence rates. Previous studies of CAM usage for cancer have found that women and people with breast cancer are more likely to seek CAM interventions. This was the case with our findings as well, with breast cancer by far the most commonly described cancer type, making up nearly a quarter of campaigns. Cancers in the female reproductive system were also much more common than their incidence or mortality rates would suggest, including ovarian and cervical cancer. At the same time, campaigns for people with prostate cancer

**Table 5. CAM treatment sought.**

| CAM Treatment | # | % |
|---|---|---|
| Supplements (teas, antioxidants) | 422 | 30.23 |
| Food/Diet (organic, nutrition) | 293 | 20.99 |
| IV Vitamin C | 276 | 19.77 |
| Naturopathy | 226 | 16.19 |
| Cannabis (CBD, RSO, THC) | 211 | 15.11 |
| Vitamins and Minerals | 206 | 14.76 |
| Herbal Remedies (mushrooms) | 159 | 11.39 |
| Oxygen Treatments (HBOT) | 146 | 10.46 |
| Immune System Boosting | 123 | 8.81 |
| Acupuncture | 121 | 8.67 |
| Detox (purges, cleanses, chelation) | 111 | 7.95 |
| Hyperthermia (heat therapy) | 110 | 7.88 |
| Juicing | 96 | 6.88 |
| Ozone Treatments | 87 | 6.23 |
| Gerson Therapy | 82 | 5.87 |
| Unspecified Alternative Treatments (Holistic, Natural) | 77 | 5.52 |
| Homeopathy | 76 | 5.44 |
| Light Treatments (Infrared, Photodynamic, Lasers) | 65 | 4.66 |
| Alternative Chemotherapy (low does, IPT) | 62 | 4.44 |
| Electromagnetic Treatments (Radio, Rife, Bemer) | 59 | 4.23 |
| Essential Oils (aromatherapy) | 52 | 3.72 |
| Mistletoe (Iscador) | 50 | 3.58 |
| Vitamin B17 (Laetrile, apricot seeds) | 50 | 3.58 |
| Energy Healing (Reiki, Qigong) | 47 | 3.37 |
| Body Work (massage) | 40 | 2.87 |
| PH Balancing (alkaline water) | 40 | 2.87 |
| Ketogenic Diet | 33 | 2.36 |
| Traditional Chinese Medicine | 27 | 1.93 |
| Chiropractic Treatment | 23 | 1.65 |
| Coffee Enemas | 23 | 1.65 |
| Dendritic Cell Therapy | 18 | 1.29 |
| Lymphatic Massage | 15 | 1.07 |
| Stem Cell Treatment | 13 | .93 |
| Faith Healing (Shamans) | 11 | .79 |
| Hypothermia (Cryotherapy, Cold Treatment) | 11 | .79 |
| Budwig Diet | 9 | .64 |
| Meditation | 9 | .64 |
| Ayurveda | 6 | .43 |
| Sonodynamic (Sound) | 6 | .43 |
| Hoxsey Protocol | 5 | .43 |
| Hydrotherapy | 5 | .36 |
| Reflexology | 5 | .36 |

were much less common than incidence and mortality rates would suggest. This provides evidence that people with breast cancer and cancers of the female reproductive system are more likely to ask for crowdfunding support for CAM cancer interventions. In addition to breast cancer, studies have identified melanoma and colorectal as common types among those seek

**Table 6. CAM provider location.**

| Provider Location | Number | Percent of Named Locations |
|---|---|---|
| Mexico | 198 | 41.86 |
| US | 169 | 35.73 |
| Germany | 37 | 7.82 |
| Canada | 19 | 4.02 |
| Thailand | 5 | 1.06 |
| UK | 5 | 1.06 |
| Spain | 4 | 0.85 |
| Turkey | 4 | 0.85 |
| Australia | 3 | 0.63 |
| Philippines | 3 | 0.63 |
| Austria | 2 | 0.43 |
| China | 2 | 0.43 |
| Czech Republic | 2 | 0.43 |
| Hungary | 2 | 0.43 |
| India | 2 | 0.43 |
| Jamaica | 2 | 0.43 |
| Japan | 2 | 0.43 |
| Latvia | 2 | 0.43 |
| New Zealand | 2 | 0.43 |
| Switzerland | 2 | 0.43 |
| Brazil | 1 | 0.21 |
| Costa Rica | 1 | 0.21 |
| Ecuador | 1 | 0.21 |
| Indonesia | 1 | 0.21 |
| Nicaragua | 1 | 0.21 |
| Poland | 1 | 0.21 |

CAM interventions. Colorectal cancer was the second most common named cancer type in our findings and both cancer types were found at rates similar to those for cancer mortality rates in the US.

Cancers of the brain were the third most common type in our findings and appeared at more than twice the mortality rate in the US. These campaigns also appeared more commonly than in a study of crowdfunding campaigns for both conventional and CAM cancer treatments. Brain cancers have generally not been discussed in connection to CAM, though there is evidence of interest in CAM modalities in people with this form of cancer in Switzerland [25]. Our findings suggest that this group warrants more exploration in relation to their interest in CAM interventions.

Lung cancer counts for nearly a quarter of cancer mortality in the US and 10.9% of cancer crowdfunding campaigns for both conventional and CAM interventions. However, only 6.0% of campaigns in this study reported a diagnosis of lung cancer. This discrepancy could be due to lower income levels and educational attainments among those with lung cancer, factors associated with lower CAM use [26]. It is also possible that stigmatized medical conditions or behaviours such as smoking appear less commonly in crowdfunding campaigns [27, 28].

Previous studies have shown that vitamins, minerals, and natural supplements are the most commonly used CAM cancer interventions. This was the case in our findings as well, with dietary supplements and dietary modifications most common. Among those seeking vitamin and

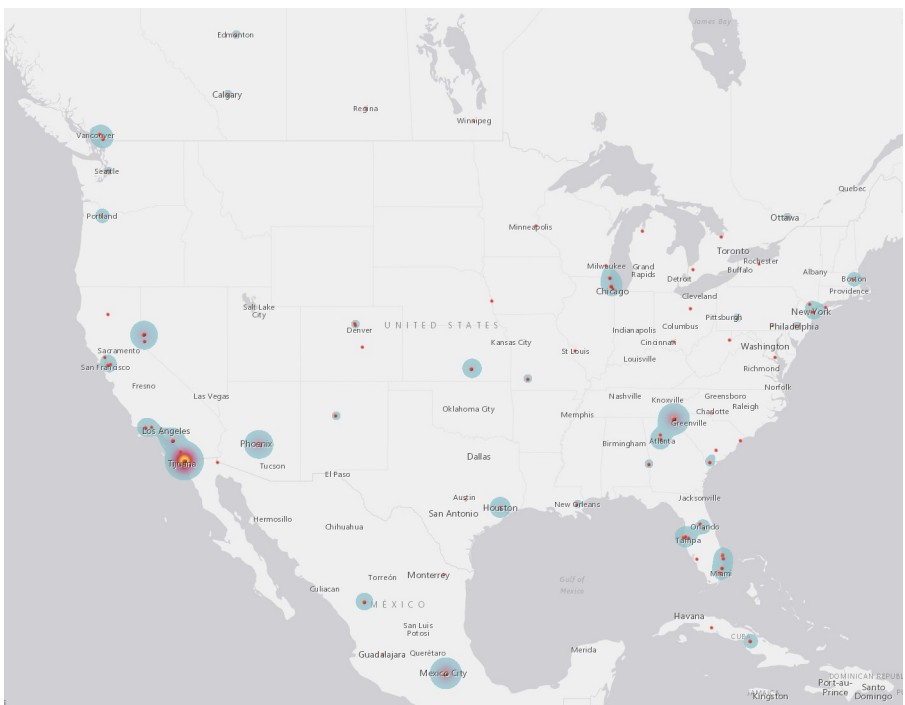

**Fig 4. North and Central America CAM providers.**

mineral supplements, high dose vitamin C, typically received intravenously, stood out as being highly sought after and, to a lesser degree, vitamin B17 or laetrile. Among supplements, cannabis products were very common, including CBD and RSO, and mistletoe or Iscador. Common CAM interventions outside of these categories included oxygen treatments and HBOT, hyperthermia, ozone treatments, and homeopathy. Gerson therapy has previously been flagged as a common and potentially dangerous CAM treatment, including in crowdfunding campaigns. This intervention was common here too. These interventions are generally not well supported by evidence and can cause negative side effects and interact with conventional treatments [4, 29].

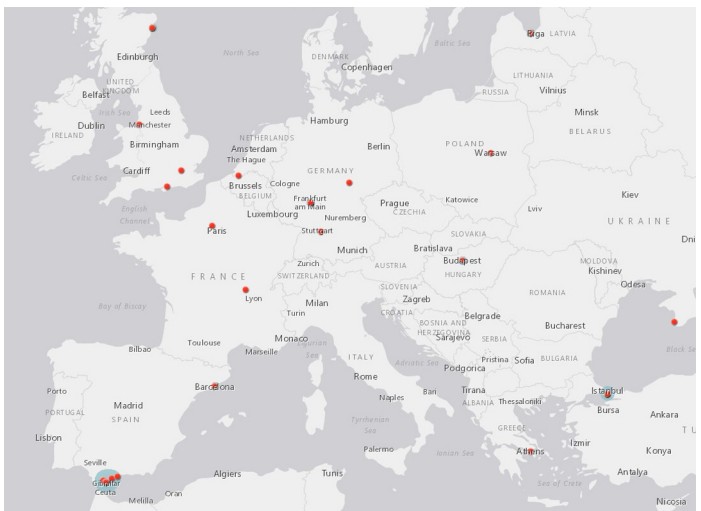

**Fig 5. European CAM providers.**

**Table 7. CAM provider name.**

| Provider Name | Number | Percent of Named Providers |
|---|---|---|
| Other | 114 | 32.02 |
| CHIPSA Hospital | 39 | 10.96 |
| Hope4Cancer | 33 | 9.27 |
| Gerson Institute | 23 | 6.46 |
| Immunity Therapy Center | 15 | 4.21 |
| Oasis of Hope | 14 | 3.93 |
| Forsythe Cancer Care Center | 11 | 3.09 |
| Cancer Center for Healing | 10 | 2.81 |
| Hoxsey Biomedical Center | 10 | 2.81 |
| Northern Baja Gerson Center | 10 | 2.81 |
| Sanoviv Medical Institute | 8 | 2.25 |
| Burzynski Clinic | 6 | 1.69 |
| Verita Life | 6 | 1.69 |
| Arcadia Praxisklinik | 5 | 1.40 |
| Block Center for Integrative Cancer Treatment | 4 | 1.12 |
| Cancer Treatment Centers of America | 4 | 1.12 |
| EuroMed Foundation | 4 | 1.12 |
| Integrated Health Clinic | 4 | 1.12 |
| Port Moody Integrated Health | 4 | 1.12 |
| Riordan Clinic | 4 | 1.12 |
| San Diego Clinic | 4 | 1.12 |
| An Oasis of Hope | 3 | 0.84 |
| Angel Farms | 3 | 0.84 |
| ChemoThermia Oncology Center | 3 | 0.84 |
| Envita Medical Center | 3 | 0.84 |
| Issels Immuno-Oncology | 3 | 0.84 |
| Namaste Health Center | 3 | 0.84 |
| Optimum Health Institute | 3 | 0.84 |
| Utopia Wellness | 3 | 0.84 |

A previous study of crowdfunding for CAM cancer interventions noted the Burzynski clinic in Texas and the Hallwang Private Oncology Clinic in Germany as common destinations for recipients. By comparison, our study found a very high concentration of destinations in Baja California and, specifically, Tijuana, including CHIPSA Hospital, Hope4Cancer, the Gerson Institute, the Immunity Therapy Center, and Oasis of Hope. These providers have been criticized as offering ineffective and potentially dangerous interventions and for misleading marketing practices [30]. While the Burzynski Clinic appeared in 1.6% of named providers, the Hallwang Clinic appeared only once.

The single appearance of the Hallwang clinic in our findings was likely due to GoFundMe's decision in March, 2019 to ban campaigns for treatment at that clinic due to concerns with whether campaigners were well informed about treatments offered there [31]. We suggest that our findings could similarly be used to restrict campaigns for treatments at facilities that have been linked to misleading information by providers or treatments with poor evidence of efficacy and increased risks to people with cancer. Similarly, our findings can and should be used to help crowdfunding platforms, clinicians, researchers, and patient advocates to identify patient groups and CAM intervention types that deserve greater research and focused interventions including education campaigns. These findings can also be used to support efforts to

further regulate the CAM sector, particularly in instances where specific interventions are being offered for specific cancer types without evidentiary support and based on misleading claims by providers. However, our findings also demonstrate the global nature of CAM provision, where clusters of providers can take advantage of international boundaries and benefit from weaker regulatory oversight to offer CAM treatments to non-nationals.

This study faces several limitations. The information provided in crowdfunding campaigns is self-reported and therefore may be incomplete or inaccurate. The campaign location is attributed to the campaign organizer, who may be a different person and in a different location than the campaign recipient. Campaigners and recipients are typically close friends and family, however, reducing the impact of this limitation on understanding recipient locations. As we captured campaign data at a single point in time, some campaigns initiated during the prior year would have been previously closed and therefore not captured in our findings. Thus, the overall number of crowdfunding campaigns for CAM cancer interventions is larger than that reported here.

As our and others' findings show, people with cancer seeking CAM interventions are largely a very ill group of people. They are highly vulnerable to the harms of lost financial resources due to ineffective interventions, negative side effects and drug interactions from some CAM interventions, encouragement to forego palliative care, and lost and exploited hope. These campaigners pass misinformation about the efficacy of CAM interventions to a wide audience through highly compelling testimonials. These findings display a growing problem in the use and funding of CAM cancer interventions and, at the same time, an opportunity for timely information about CAM usage and targeted interventions where justified.

## Supporting information

**S1 Appendix. CAM search terms.**
(DOCX)

## Acknowledgments

Patients or the public were not involved in the design, or conduct, or reporting, or dissemination plans of our research.

## Author Contributions

**Conceptualization:** Jeremy Snyder, Marco Zenone, Timothy Caulfield.

**Data curation:** Jeremy Snyder, Marco Zenone.

**Formal analysis:** Jeremy Snyder, Marco Zenone, Timothy Caulfield.

**Methodology:** Jeremy Snyder, Marco Zenone, Timothy Caulfield.

**Supervision:** Jeremy Snyder.

**Visualization:** Jeremy Snyder.

**Writing – original draft:** Jeremy Snyder, Marco Zenone.

**Writing – review & editing:** Jeremy Snyder, Marco Zenone, Timothy Caulfield.

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
