## [Decision Letter · Decision Letter 0]

9 Oct 2020

PONE-D-20-09717

Crowdfunding for Complementary and Alternative Medicine: What are Cancer Patients Seeking?

PLOS ONE

Dear Dr. Snyder,

Thank you for submitting your manuscript to PLOS ONE. After careful consideration, we feel that it has merit but does not fully meet PLOS ONE’s publication criteria as it currently stands. Therefore, we invite you to submit a revised version of the manuscript that addresses the points raised during the review process.

The reviewers were highly positive about the focus and conduct of the study. In responding to their comments, I encourage you specifically to attend to:

- the abstract and discussion foreground findings related to gender, socio-economic status, and educational attainment, but these are present in your results only implicitly; consider presenting analyses involving demographic data on the recipients of crowdfunding campaigns more explicitly or editing the discussion to be consistent with the findings of this study

- consider how you might add qualitative dimensions to the results; given that campaigns are publicly available on the internet, it may not be appropriate to provide direct quotations (which are then identifying), but perhaps illustrative examples or composite sketches could give readers a richer picture of the campaigns you describe

- consider re-organizing the results section so that there is little duplication between the text and the tables

We look forward to receiving your revised manuscript.

Kind regards,

Quinn Grundy, PhD, RN

Academic Editor

PLOS ONE

Additional Editor Comments:

Table 8 appears to be the appendix you reference in the Methods section - please re-label or cite Table 8 in the text.

Journal Requirements:

"The research was supported by a grant from the Greenwall Foundation. Patients or the public

were not involved in the design, or conduct, or reporting, or dissemination plans of our research."

"The authors received no specific funding for this work."

4. Please ensure that you refer to Figures 1-5 in your text as, if accepted, production will need this reference to link the reader to the figures.

5. We note that Figures 1-5 in your submission contain map images which may be copyrighted. All PLOS content is published under the Creative Commons Attribution License (CC BY 4.0), which means that the manuscript, images, and Supporting Information files will be freely available online, and any third party is permitted to access, download, copy, distribute, and use these materials in any way, even commercially, with proper attribution. For these reasons, we cannot publish previously copyrighted maps or satellite images created using proprietary data, such as Google software (Google Maps, Street View, and Earth). For more information, see our copyright guidelines: http://journals.plos.org/plosone/s/licenses-and-copyright.

5.1.    You may seek permission from the original copyright holder of Figures 1-5 to publish the content specifically under the CC BY 4.0 license. 

5.2.    If you are unable to obtain permission from the original copyright holder to publish these figures under the CC BY 4.0 license or if the copyright holder’s requirements are incompatible with the CC BY 4.0 license, please either i) remove the figure or ii) supply a replacement figure that complies with the CC BY 4.0 license. Please check copyright information on all replacement figures and update the figure caption with source information. If applicable, please specify in the figure caption text when a figure is similar but not identical to the original image and is therefore for illustrative purposes only.

Reviewers' comments:

Reviewer's Responses to Questions

**Comments to the Author**

1. Is the manuscript technically sound, and do the data support the conclusions?

Reviewer #1: Yes

Reviewer #2: Yes

2. Has the statistical analysis been performed appropriately and rigorously? 

Reviewer #1: Yes

Reviewer #2: Yes

3. Have the authors made all data underlying the findings in their manuscript fully available?

Reviewer #1: Yes

Reviewer #2: Yes

4. Is the manuscript presented in an intelligible fashion and written in standard English?

Reviewer #1: Yes

Reviewer #2: Yes

5. Review Comments to the Author

Reviewer #1: Well designed and executed study delineating ongoing trends on CAM cancer fundraising activities on GoFundMe. Despite strong media coverage of the subject and GoFundMe's awareness of the problem (including banning one alternative cancer clinic in March 2019), GoFundMe remains a hotbed of economic activity surrounding dubious healthcare. We should have strong concern that people with advanced cancers and their caregivers are being taken advantage of. As the authors note the interest surrounding many of these therapies warrants greater investigation for each. Patients and families are spending time and energy and investing hope into each of these purported treatments. The medical community needs to do more than respond with dismay. There is a rich amount of information here about trends of fundraising activity. I would be interested in the fundraising totals allocated as well but understand the study is meant to focus on expressed interests of patients/caregivers not the success of the campaigns.

Reviewer #2: This is an interesting and important piece about a specific type of healthcare (CAM) and its prevalence on GoFundMe. My first reaction was “no literature review?” but I ended up kind of liking the “let’s get to it” style of the paper, which I think is common on PONE. It is well-written, based on solid research, and offers important commentary at the intersection of crowdfunding, cancer, and healthcare. I would recommend that this piece be accepted, with some revisions. My few suggestions below are meant to improve what is already a strong paper.

• It would be helpful to have a bit more information on who is pursuing CAM and by extension, who is not. At least a breakdown of gender of subjects? I am surprised the authors did not code for visible minority status when they were looking through pages. Why not? I’d at least explain.

• A few questions on methods: Why June 4 to June 4? It feels arbitrary? Are there any intercoder reliability statistics on the coding of inclusion/exclusion? Some more clarity on the geographic parameters of the data (or lack thereof) would be helpful—was it just a case of winnowing the global set of cases? It is not clear as written.

• Pages 8 and 9 were a slog—I’d rely solely on tables with language after each one—this was impossible to work through.

• I did wonder at the lack of texture and depth in the data—why not use some of the qualitative data (text, image description) to bolster and enhance the arguments? Why not let some of these subjects speak for themselves? I’m not suggesting a massive infusion of qualitative data, but it would be a stronger paper if the subject voices could be heard.

• The data on gender, education, and income are not featured as strongly as one might expect, given their featuring in the abstract. I’d add a bit more to this discussion.

• There is some analytical slippage on page 10 when the authors talk of a “gendered dimension”—the numbers are not evidence of CAM usage, with gender variation. They are evidence of variation in asking for money for CAM treatments. It may be the case that men are not asking as much as women, but this is a different point.

• Finally, I would encourage the authors to move beyond “perhaps we should close some of these bad clinics” thinking to consider the bigger picture. Maybe they are providing more evidence that the whole enterprise stinks? There is already plenty of data showing that crowdfunding is inequitable. Wouldn’t it be better to have all this regulated and funded by the state? A discussion of this point would make the paper more wide-reaching in scope

In sum, it’s a very nice paper and I applaud the authors for their efforts. Looking forward to seeing it in print!

6. PLOS authors have the option to publish the peer review history of their article (what does this mean?). If published, this will include your full peer review and any attached files.

Reviewer #1: **Yes: **Ford Vox, MD

Reviewer #2: No

---

## [Author Response · Author response to Decision Letter 0]

19 Oct 2020

Thank you for the opportunity to revise our manuscript and to the reviewers and editor for their very helpful comments and suggestions. All comments are reproduced below in full and responses follow each ccomment.

- the abstract and discussion foreground findings related to gender, socio-economic status, and educational attainment, but these are present in your results only implicitly; consider presenting analyses involving demographic data on the recipients of crowdfunding campaigns more explicitly or editing the discussion to be consistent with the findings of this study

We have edited the abstract to be more in line with the discussion.

- consider how you might add qualitative dimensions to the results; given that campaigns are publicly available on the internet, it may not be appropriate to provide direct quotations (which are then identifying), but perhaps illustrative examples or composite sketches could give readers a richer picture of the campaigns you describe

We have now added characteristic quotations from the campaigns throughout the results section as per reviewer two’s suggestion.

- consider re-organizing the results section so that there is little duplication between the text and the tables

We now use quotes from the campaigns to better characterize the quantitative data that is presented in the results and tables.

Table 8 appears to be the appendix you reference in the Methods section - please re-label or cite Table 8 in the text.

We have re-labeled this table as Appendix

"The research was supported by a grant from the Greenwall Foundation. Patients or the public were not involved in the design, or conduct, or reporting, or dissemination plans of our research."

"The authors received no specific funding for this work."

We would like to amend the funding statement to indicate that “The authors received funding from the Greenwall Foundation that supported this work.”

Data is available at this link: https://researchdata.sfu.ca/pydio_public/b8bc37

4. Please ensure that you refer to Figures 1-5 in your text as, if accepted, production will need this reference to link the reader to the figures.

All figures are now referred to in the text.

5. We note that Figures 1-5 in your submission contain map images which may be copyrighted. All PLOS content is published under the Creative Commons Attribution License (CC BY 4.0), which means that the manuscript, images, and Supporting Information files will be freely available online, and any third party is permitted to access, download, copy, distribute, and use these materials in any way, even commercially, with proper attribution. For these reasons, we cannot publish previously copyrighted maps or satellite images created using proprietary data, such as Google software (Google Maps, Street View, and Earth). For more information, see our copyright guidelines: http://journals.plos.org/plosone/s/licenses-and-copyright.

We now use figures drawn from Arcgis: https://doc.arcgis.com/en/arcgis-online/reference/access-use-constraints.htm

Reviewer #1: Well designed and executed study delineating ongoing trends on CAM cancer fundraising activities on GoFundMe. Despite strong media coverage of the subject and GoFundMe's awareness of the problem (including banning one alternative cancer clinic in March 2019), GoFundMe remains a hotbed of economic activity surrounding dubious healthcare. We should have strong concern that people with advanced cancers and their caregivers are being taken advantage of. As the authors note the interest surrounding many of these therapies warrants greater investigation for each. Patients and families are spending time and energy and investing hope into each of these purported treatments. The medical community needs to do more than respond with dismay. There is a rich amount of information here about trends of fundraising activity. I would be interested in the fundraising totals allocated as well but understand the study is meant to focus on expressed interests of patients/caregivers not the success of the campaigns.

We agree that this information is of interest and so have added the total and median funding requests and pledges to the first paragraph of the findings section, namely: “They requested $39,611,973.20 (median $19,880) and were pledged $12,756,563.92 (median $5,055.50) or 32.2% of that amount”.

Reviewer #2: This is an interesting and important piece about a specific type of healthcare (CAM) and its prevalence on GoFundMe. My first reaction was “no literature review?” but I ended up kind of liking the “let’s get to it” style of the paper, which I think is common on PONE. It is well-written, based on solid research, and offers important commentary at the intersection of crowdfunding, cancer, and healthcare. I would recommend that this piece be accepted, with some revisions. My few suggestions below are meant to improve what is already a strong paper.

• It would be helpful to have a bit more information on who is pursuing CAM and by extension, who is not. At least a breakdown of gender of subjects? I am surprised the authors did not code for visible minority status when they were looking through pages. Why not? I’d at least explain.

We agree that this would be valuable context to add to the study. We did not do so in this case because we aimed for a big picture sense of the market for CAM cancer treatments and because the database we used does not capture images – only campaign text. This is now clarified in the methods section. We think that it would be worthwhile to take a smaller dataset of campaigns for CAM cancer treatment and assess gender and visible minority status. We aim to do so in the future.

• A few questions on methods: Why June 4 to June 4? It feels arbitrary? Are there any intercoder reliability statistics on the coding of inclusion/exclusion? Some more clarity on the geographic parameters of the data (or lack thereof) would be helpful—was it just a case of winnowing the global set of cases? It is not clear as written.

June 4 was the start date for our search process and that time range was selected to capture a sample of campaigns created over the most recent one-year period. There were no geographic limits placed on the sample, only the temporal limit. The third author reviewed 5% of campaign codes as they were added by the first and second authors and the team discussed and resolved divergences in coding as they arose. We did not gather intercoder reliability statistics.

• Pages 8 and 9 were a slog—I’d rely solely on tables with language after each one—this was impossible to work through.

We have added qualitative data to the results section that breaks up the presentation of the quantitative data and improves the readability of this section.

• I did wonder at the lack of texture and depth in the data—why not use some of the qualitative data (text, image description) to bolster and enhance the arguments? Why not let some of these subjects speak for themselves? I’m not suggesting a massive infusion of qualitative data, but it would be a stronger paper if the subject voices could be heard.

Thank you for this suggestion. We have now added quotes from the campaigns that gives characteristic language from the campaigners.

• The data on gender, education, and income are not featured as strongly as one might expect, given their featuring in the abstract. I’d add a bit more to this discussion.

We have modified the abstract to be more consistent with the discussion.

• There is some analytical slippage on page 10 when the authors talk of a “gendered dimension”—the numbers are not evidence of CAM usage, with gender variation. They are evidence of variation in asking for money for CAM treatments. It may be the case that men are not asking as much as women, but this is a different point.

We now are more precise with this claim, specifically that it supports greater fundraising interest by women for CAM cancer interventions.

• Finally, I would encourage the authors to move beyond “perhaps we should close some of these bad clinics” thinking to consider the bigger picture. Maybe they are providing more evidence that the whole enterprise stinks? There is already plenty of data showing that crowdfunding is inequitable. Wouldn’t it be better to have all this regulated and funded by the state? A discussion of this point would make the paper more wide-reaching in scope.

Thank you for raising this point. While we agree that crowdfunding is on the whole deeply problematic and inequitable, we think it is important to distinguish between particularly problematic practices such as crowdfunding for unproven and potentially dangerous or worthless medical interventions and crowdfunding for, say, a Black Lives Matter group or wait staff put out of work by the COVID-19 pandemic. We agree that states should regulate CAM treatments but, as these campaigns show, the global nature of the marketplace makes this difficult. If the US or Canada have stricter regulations, campaigners may seek funding to go to Tijuana, which our findings show is now happening. We have added to the discussion to make this point clear. We do not think that states should be funding CAM cancer treatments unless there is clear evidence of efficacy and safety.

---

## [Editor Report · Decision Letter 1]

27 Oct 2020

Crowdfunding for Complementary and Alternative Medicine: What are Cancer Patients Seeking?

PONE-D-20-09717R1

Dear Dr. Snyder,

We’re pleased to inform you that your manuscript has been judged scientifically suitable for publication and will be formally accepted for publication once it meets all outstanding technical requirements.

Kind regards,

Quinn Grundy, PhD, RN

Academic Editor

PLOS ONE
---

## [Editor Report · Acceptance letter]

10 Nov 2020

PONE-D-20-09717R1 

Crowdfunding for Complementary and Alternative Medicine: What are Cancer Patients Seeking? 

Dear Dr. Snyder:

I'm pleased to inform you that your manuscript has been deemed suitable for publication in PLOS ONE. Congratulations! Your manuscript is now with our production department. 

Kind regards, 

on behalf of

Dr. Quinn Grundy 

Academic Editor

PLOS ONE